# Hollow Mesoporous CeO_2_-Based Nanoenzymes Fabrication for Effective Synergistic Eradication of Malignant Breast Cancer via Photothermal–Chemodynamic Therapy

**DOI:** 10.3390/pharmaceutics14081717

**Published:** 2022-08-17

**Authors:** Huaxin Tan, Yongzhen Li, Jiaying Ma, Peiyuan Wang, Qiaoling Chen, Lidan Hu

**Affiliations:** 1Department of Biochemistry and Molecular Biology, The Key Laboratory of Ecological Environment and Critical Human Diseases Prevention of Hunan Province Department of Education, School of Basic Medicine, Hengyang Medical School, University of South China, Hengyang 421001, China; 2Department of Anesthesiology, The First Affiliated Hospital of Xiamen University, Xiamen 361001, China; 3Key Laboratory of Design and Assembly of Functional Nanostructures, Fujian Institute of Research on the Structure of Matter, Chinese Academy of Sciences, Fuzhou 350002, China

**Keywords:** hollow CeO_2_-based nanoenzyme, ICG, chemodynamic therapy enhancement, photothermal therapy, synergistic effect

## Abstract

CeO_2_-based nanoenzymes present a very promising paradigm in cancerous therapy, as H_2_O_2_ can be effectively decomposed under the electron transmit between Ce^3+^ and Ce^4+^. However, the limitations of endogenous H_2_O_2_ and intracellular low Fenton-like reaction rate lead to single unsatisfied chemodynamic therapy (CDT) efficacy. Other therapeutic modalities combined with chemodynamic therapy are generally used to enhance the tumor eradiation efficacy. Here, we have synthesized a novel hollow pH-sensitive CeO_2_ nanoenzyme after a cavity is loaded with indocyanine green (ICG), as well as with surface modification of tumor targeting peptides, Arg-Gly-Asp (denoted as HCeO_2_@ICG-RGD), to successfully target tumor cells via α_v_β_3_ recognition. Importantly, in comparison with single chemodynamic therapy, a large amount of reactive oxygen species in cytoplasm were induced by enhanced chemodynamic therapy with photothermal therapy (PTT). Furthermore, tumor cells were efficiently killed by a combination of photothermal and chemodynamic therapy, revealing that synergistic therapy was successfully constructed. This is mainly due to the precise delivery of ICG and release after HCeO_2_ decomposition in cytoplasm, in which effective hyperthermia generation was found under 808 nm laser irradiation. Meanwhile, our HCeO_2_@ICG-RGD can act as a fluorescent imaging contrast agent for an evaluation of tumor tissue targeting capability in vivo. Finally, we found that almost all tumors in HCeO_2_@ICG-RGD+laser groups were completely eradicated in breast cancer bearing mice, further proving the effective synergistic effect in vivo. Therefore, our novel CeO_2_-based PTT agents provide a proof-of-concept argumentation of tumor-precise multi-mode therapies in preclinical applications.

## 1. Introduction

Similar to toxic substances, reactive oxygen species (ROS) can induce tumor cell apoptosis by destroying biological molecules, such as functional protein or DNA damage [1,2,3]. Chemodynamic therapy (CDT), as the most well-known ROS-based anticancer strategy, has attracted widespread attention. Compared with some oxygen-dependent therapeutic modalities, such as photodynamic therapy (PDT) under laser irradiation or sonodynamic therapy with untrasound-assistance [4,5,6], CDT could exert a significant antitumor capability under the hypoxia tumor microenvironment without any other external facilities [7]. Frequently, CDT employs catalysts, such as metal-based nanomaterials to initiate Fenton-like reactions, which are capable of converting intracellular hydrogen peroxide (H_2_O_2_) into toxic oxidative hydroxyl radicals (•OH). Consequently, this leads to tumor cell apoptosis [8]. In addition, CDT ensures normal cell safety by suppressing Fenton-like reactions with insufficient H_2_O_2_ under normal physiological microenvironment [9]. In recent years, nanoenzymes as nanocatalytic therapeutics, have been extensively used for antitumor applications, owing to their parallel catalytic activity toward natural enzymes and unique properties [10,11,12]. Numerous nanomaterials have been explored as nanoenzymes with various sizes and functional groups on the surface, which showed predominant potential in tumor eradication [13,14,15,16,17]. Among these nanoenzymes, cerium oxide (CeO_2_) nanomaterials can be applied as candidates for the development of tumor therapeutic nanoplatforms due to their catalase and peroxidase activity toward high H_2_O_2_ in tumor microenvironment at various pH values [18]. Interestingly, several studies recently discovered that H_2_O_2_ can absorb onto the surface of CeO_2_ nanocatalyst, in which a stable CeO_2_–H_2_O_2_ complex can be formed [19,20]. Specifically, under acidic microenvironment, the coordinated H_2_O_2_ molecules undergo a disproportionation procedure to yield two hydroxyl radical generation rates with overwhelming cytotoxicity against cancerous cells. Moreover, this process can be regarded as CDT [9,21,22,23]. Accordingly, CeO_2_-based nanoparticles are the rational choice in the construction of CDT-based tumor therapeutic nanoplatforms. However, to date, owing to the restriction from the total amount of H_2_O_2_ in cytoplasm of tumor cells, CDT is still not able to obtain satisfactory therapeutic performance. Therefore, improving the efficacy of CDT is the main challenge during the clinical translation of CDT nanoplatforms.

To improve the tumor therapeutic effect of CDT, combining other therapies to achieve the synergistic therapy is a relatively ideal choice. Generally, CDT nanoplatforms are designed based on metal oxide nanoparticles. This design is very convenient for loading or anchoring other therapeutics to achieve combination therapy. Similar to anticancer drugs, camptothecin can be successfully loaded into Fe-based metal organic frameworks (MOF) to combine CDT with chemotherapy [24]. Specifically, indocyanine green (ICG) has been reported to be an effective NIR light absorber for laser-mediated photothermal therapy, that has been approved by the Food and Drug Administration (FDA) [25,26,27]. Ce doped Cu–Al layered double hydroxide ultrathin nanosheets with indocyanine-green (ICG)-loaded nanoparticles were capable of generating effective hyperthermia, which substantially heightens the Fenton-like reaction of Cu^+^ and Ce^3+^ with local H_2_O_2_ [28]. Interestingly, an increased local tumor temperature induced by PTT could result in the generation of hydroxyl radicals. Meanwhile, the production of a large amount of toxic hydroxyl radicals further delays the heat shock protein expression, which leads the cancerous cells to be more sensitive toward heat [29,30,31,32,33]. Therefore, the synergistic enhancement of PTT+CDT in tumor therapy could be achieved. In this context, the co-delivery of ICG nanosystem through the CeO_2_ nanoenzyme is expected to achieve specific and enhanced therapeutic effects.

Based on the above investigations, we herein demonstrate the successful fabrication of a novel hollow CeO_2_ nanoenzyme with ICG loading (HCeO_2_@ICG). This can serve as the active therapeutic nanoenzyme for CDT via Ce^3+^/Ce^4+^ under acidic microenvironment and Fenton-like reation ehancement by PTT, as illustrated in Figure 1. After modification of NH_2_ groups on the surface of tumor cell membrane α_v_β_3_ targted peptides, RGD molecules were successfully anchored on the surface of HCeO_2_@ICG (HCeO_2_@ICG-RGD). This nanoenzyme could be effectively targeted in tumor tissues and degraded in lysosome for ICG release. Furthermore, the degraded CeO_2_ can be used for ROS generation. Importantly, hyperthermia generated under 808 nm laser irradiation further facailited intracellular ROS production. After tail vein injection, HCeO_2_@ICG-RGD can be used as a contrasting agent in the second near-infrared region window (NIR II, 900–1700 nm). It accumulates in solid tumor at 24 h post-injection, under the observation of NIR II fluorescent imaging, then after 808 nm laser irradiation for 5 min, malignant tumors were completely eradicated. These findings demonstrate the great potential of ICG-loaded CeO_2_-based hollow nanoenzyme for clinical translation of malignant breast cancer inhibition.

## 2. Materials and Methods

### 2.1. Virus-like Mesoporous Silica Synthesis

The virus-like mesoporous silica template was obtained by a special epitaxial growth approach with the hexadecyltrimethylammonium (CTAB) as the template, tetraethyl orthosilicate (TEOS) as a Si precursor, NaOH as the reductant, and cyclohexane as the oil solvent [34]. Typically, 1.0 g of CTAB and 0.8 mL of NaOH (0.1 M) were first added to 50 mL of H_2_O_2_ and stirred in oil bath at 60 °C for 30 min. Then, 20 mL of 20% of TEOS/cyclohexane mixture was added to the water layer and stirring was maintained in oil bath at 60 °C for 2 days. Thereafter, the as-made products were alternatively washed 6 times by water and ethanol, followed by centrifugation at 13,000 rpm for 10 min. Finally, the virus-like mesoporous nanoparticles were dispersed in 50 mL of acetone and refluxed at 60 °C in oil bath for more than 12 h to remove the residual surfactants. The final silica samples were alternatively washed by water and ethanol, followed by drying under vacuum overnight at 45 °C.

### 2.2. HCeO_2_@ICG Synthesis

For HCeO_2_@ICG synthesis, 100 mg of virus-like silica templates and Ce(NO_3_)_3_·6H_2_O (0.1232 g) were first co-dissolved in 50 mL of deionized water and moderately stirred for 30 min at 90 °C in oil bath. Then, 0.105 g of hexamethylenetetramine power was added to this mixture, and the solution was vigorously stirred for 120 min at 90 °C in oil bath. The final core-shell samples (VmSi@CeO_2_) were washed three times by deionized water, followed by centrifugation at 5000 rpm for 5 min and oven dried for 12 h at 60 °C. Finally, the hard template, virus-like mesoporous silica was etched using 60 mL of 0.1 M Na_2_CO_3_, stirred at 75 °C in oil bath for 12 h, and washed more than three times with deionized water. After oven-drying (60 °C) the mixture overnight, hollow mesoporous cerium oxide nanoparticles with rough surface (HCeO_2_) were obtained.

Furthermore, 1 mg of ICG was dissolved in ethanol (2.0 mL). Then, 25 mg of HCeO_2_ was added to the above mixture and the solution was continuously stirred for 48 h under room temperature. The ICG molecules could be successfully adsorbed onto the stacking mesopores and the hollow cavity of HCeO_2_. The as-made ICG-loaded HCeO_2_ nanoenzymes (HCeO_2_@ICG) were washed three times, followed by by centrifugation at 5000 rpm for 10 min. The precise loading of ICG content in HCeO_2_@ICG was quantified from a standard curve by measuring the UV absorbance.

### 2.3. HCeO_2_@ICG-RGD Synthesis

For HCeO_2_@ICG-RGD synthesis, 100 mg of HCeO_2_@ICG and 20 mg of DSPE-PEG_2000_-NH_2_ were first co-dissolved in 5 mL of chloroform. Then, the amino group modified HCeO_2_@ICG (HCeO_2_@ICG-PEG) was obtained after the chloroform was all slowly evaporated in a fume cupboard. After the removal of residual DSPE-PEG_2000_-NH_2_ by ultra-centrifugation at 15,000 rpm for 30 min, the as-made HCeO_2_@ICG-PEG was acquired and re-dispersed in deionized water.

Subsequently, HCeO_2_@ICG-RGD was prepared. First, 5 mg of RGD was dissolved in 5 mL of 2-(N-morpholino) ethanesulfonic acid (MES) buffer (0.1 M, pH = 5.5). Then, 20 mg of N-(3-dimethylaminopropyl)-N′-ethylcarbodiimide hydrochloride (EDC) and 20 mg of N-hydroxysuccinimide (NHS) were added and gently stirred for 120 min at room temperature. Thereafter, 50 mg of HCeO_2_@ICG was added into the above mixture and continuously stirred for 12 h. The as-prepared RGD anchored HCeO_2_@ICG was washed by deionized water, followed by centrifugation at 5000 rpm for 10 min. The final product, HCeO_2_@ICG-RGD, was dispersed in deionized water.

### 2.4. Cell Targeting and Intracellular ROS Generation Studies

The cellular uptake of HCeO_2_@ICG-RGD and HCeO_2_@ICG was observed by confocal laser scanning microscopy (CLSM). Briefly, 4T1 breast tumor cells were first seeded into 6-well plates with a cellular density of 1 × 10^5^ per well and cultured for 24 h under 1 mL of 1640 medium (10% FBS, 100 units/mL of penicillin, and 100 μg/mL of streptomycin). After the treatment with HCeO_2_@ICG-RGD and HCeO_2_@ICG at a final concentration of 200 μg/mL for 12 h, respectively, all of the cell samples were washed with 1× PBS more than three times to stain the nuclei. Then, 1 μg/mL of DAPI was added into the above cells for 0.5 h. The as-prepared cell samples were observed under CLSM with 488 nm light excitation and an observation window from 550–600 nm.

The intracellular ROS generated evaluation of HCeO_2_@ICG-RGD was observed by CLSM. Briefly, 1 × 10^5^ per well of 4T1 cells were seeded into 6-well plates and cultured for 24 h under 1 mL of 1640 medium (10% FBS, 100 units/mL of penicillin, and 100 μg/mL of streptomycin). After the incubation with 200 μg/mL HCeO_2_@ICG-RGD for 12 h, 808 nm laser was irradiated for 5 min (0.75 W cm^−2^). Then, 4T1 cells co-incubated with PBS or HCeO_2_@ICG-RGD were prepared following the same procedure. Furthermore, intracellular ROS production of all groups were estimated by DCHF-DA staining.

### 2.5. Fluorescent Imaging of Tumor Targeting and Photothermal Imaging In Vivo

All animal investigations were approved by the Medical Ethics Committee in South China University (USC202205XS02), which were performed in accordance with all relevant guidelines (Hengyang, China). Herein, 6-week-old female BALB/c nude mice were purchased from Shanghai SLAC Laboratory Animal Co., Ltd. (Shanghai, China). Subcutaneous tumor-bearing nude mice models were prepared by transcutaneous injection of tumor cells (1 × 10^6^ 4T1 cells dispersed in 150 μL PBS) into the right hindlimb muscle. When the tumor volume increased to ~200 mm^3^, the ICG and HCeO_2_@ICG-RGD were intravenously injected into tumor-bearing mice, respectively. Then, all the mice were imaged by the NIR fluorescent imaging system (NIR-OPTICS Series III 900/1700, 1000 nm long-pass filter) in a range of post-injection durations. Thereafter, major organs and tumors in the above two groups were collected after 24 h post-injection to obtain an ex vivo NIR II fluorescent image. Infrared region (IR) thermal imaging was employed for photothermal conversion in vivo. Tumor-bearing BALB/c nude mice received tail vein injection of 1X PBS and HCeO_2_@ICG-RGD, respectively. The tumor site was irradiated with an 808 nm laser (1 W cm^−2^) for 5 min at 24 h post-injection. After laser irradiation, in vivo IR thermal images were immediately obtained by a NIR thermal camera (FOTRIC 225s, Chinaprotech, Shanghai, China).

### 2.6. Synergistic Antitumor Studies of HCeO_2_@ICG-RGD

To investigate antitumor ablation, tumor-bearing BALB/c nude mice were divided into five groups (n = 3) when the subcutaneous tumors reached a volume of ~150 mm^3^. The animals had received various intravenous injections (PBS, HCeO_2_+laser, HCeO_2_-RGD+laser, HCeO_2_@ICG+laser, and HCeO_2_@ICG-RGD+laser) with the same HCeO_2_ concentration at 8.2 mg/kg. Twenty-four hours after injection with the above formulations, nude mice in the last four groups were subjected to 5 min of NIR laser irradiation (808 nm, 1 W cm^−2^). Tumor volume and body weight were recorded every 3 days. Importantly, tumors after the 9-day treatment were dissected and cut into tissue sections of 10 μm in thickness. Hematoxylin and eosin (H&E) and TdT-mediated dTUP-biotin nick end-labeling (TUNEL) staining studies were subsequently conducted. Meanwhile, after the 18-day treatment, the main organs (heart, liver, spleen, lung, and kidney) were also stained by H&E.

## 3. Results and Discussion

### 3.1. HCeO_2_@ICG-RGD Fabrication

The synthesis of hollow mesoporous cerium oxide nanoparticles HCeO_2_ and final ICG loading and RGD modification are shown in Figure 1. As depicted, HCeO_2_ nanoenzymes were constructed by a hard-template strategy. Briefly, the template with ~100 nm of virus-like mesoporous silica (VmSiO_2_) nanoparticles was first synthesized by a novel biphasic (water/cyclohexane) strategy with CTAB as the surfactant, TEOS as the silica precursor, and NaOH as the reductant. VmSiO_2_ could be successfully synthesized after a 72-h reaction at the water–oil interface. Transmission electron microscope (TEM) was employed to observe the morphology of nanomaterials. As shown in Figure 2A, the as-made VmSiO_2_ presented distinct nanotubes on the mesoporous surface with a uniform and monodispersed nanostructure. Then, ~21 nm of CeO_2_ layer was in situ deposited on VmSiO_2_ under methenamine solution with Ce(NO_3_)_3_·6H_2_O as the CeO_2_-based precursor (Figure 2B). Subsequently, the inner VmSiO_2_ was removed under alkaline condition and the ~120 nm hollow cavity of CeO_2_ was well maintained without collapse (Figure 2C). According to our previous reports, nanogranules of CeO_2_ were reduced in situ, and then they assembled into a virus-like morphology [35,36]. The hollow CeO_2_ nanoshells simulated the VmSiO_2_ morphology, while the stacking mesopores of these CeO_2_-based building blocks made them promising candidates for drug delivery. Furthermore, the stacking force in hollow metal oxide nanoparticles demonstrated vulnerability under acidic microenvironment (pH = 5.6). Then, the ICG was encapsulated into the hollow cavity and mesopores of this novel pH-sensitive HCeO_2_ under room temperature (HCeO_2_@ICG). The loading efficiency was calculated as 22.2%, proving the advantage of HCeO_2_ for small therapeutics delivery. To facilitate the following surface modification and prolong the blood circulation time, DSPE-PEG_2000_-NH_2_ molecules were coated onto the HCeO_2_ surface. Finally, RGD was functionalized on HCeO_2_ via EDC–NHS reaction (HCeO_2_@ICG-RGD). As shown in Appendix A, the peak of 1423 cm^−1^ in Fourier transform infrared spectra (FTIR) was attributed to C–N from ICG, and then it disappeared, due to DSPE-PEG_2000_-NH_2_ modification. Interestingly, due to the formation of amide bond between HCeO_2_@ICG-NH_2_ and RGD, it clearly reappeared once again. Simultaneously, as displayed in X-ray diffraction (XRD) results, after ICG loading and RGD surface modification, a single broad diffused peak was still maintained in HCeO_2_@ICG-RGD, suggesting that the nanogranules-packed HCeO_2_ nanoenzymes are more susceptible to degradation (Appendix A). At the same time, we found that the specific surface area of HCeO_2_@ICG-RGD decreased to 41.6% of the original HCeO_2_, due to the cavity loading of ICG, and surface anchoring of both DSPE-PEG_2000_-NH_2_ and RGD (Appendix A). TEMs image of HCeO_2_@ICG-RGD were obtained and the hollow nanostructure with rough surface was found, revealing that the morphology of HCeO_2_ was not affected during ICG loading, NH_2_ groups modification, and RGD anchoring (Figure 2D). To further confirm the rough surface, scanning electron microscope (SEM) was performed. Clearly, homogeneous nanotubes can be detected (Figure 2F). Undoubtedly, the nanotubes are capable of improving cell membrane adhesion, which fundamentally improves cellular internalization. Meanwhile, according to dark-field and high-angle annular dark field (HAADF) images, the cavity and rough surface could also be evidently detected (Figure 2F,G). In addition, the elemental mapping results presented that Ce and O were uniformly dispersed in the shell of HCeO_2_@ICG-RGD. No other C element was found, further suggesting that after HCeO_2_ was successfully obtained, it could be used as a CDT nanoagent.

### 3.2. HCeO_2_@ICG-RGD Characterization

To further prove the ICG loading and RGD anchoring in HCeO_2_, zeta-potential values of the above four kinds of nanoenzymes were also evaluated. As displayed in Figure 3A, the zeta potential of as-prepared HCeO_2_@ICG decreased to −22.5 mV and it changed to positive charges after DSPE-PEG_2000_-NH_2_ coating. Finally, it slightly decreased after RGD functionalization (Figure 3A). All of the data suggested that ICG encapsulation in hollow cavity, NH_2_ groups, and RGD molecules modification on the nanoenzyme surface have been successfully accomplished. Then, the average size of HCeO_2_@ICG-RGD redispersed in different buffers was analyzed by dynamic light scattering (DLS). It exhibited values of ~120, 125, and 130 nm even after the 7-day incubation period in PBS, FBS, and serum, respectively, indicating the excellent size stability in biological fluids (Figure 3B). Accordingly, we subsequently studied the Ce ions and ICG release performance under acidic microenvironment. HCeO_2_@ICG-RGD was treated with buffers under pH = 5.6 or pH = 7.4. Moreover, Ce ions in the residual HCeO_2_@ICG-RGD were measured by ICP-OES. As shown in Figure 3C, they dramatically declined to 7.5% after 4 h of incubation. However, in pH = 7.4 treated solution, little Ce ions fluctuation was found and more than 90% of Ce was found in precipitation even at 4 h of co-culture. These data inspired us to investigate the morphology of HCeO_2_@ICG-RGD under different pH values. Significantly, in pH = 5.6 buffer treated group, the CeO_2_ shell became thinner after 1 h of incubation and almost all hollow nanostructures were destroyed after 2 h of treatment. Importantly, all HCeO_2_ were completely collapsed and some CeO_2_ nanogranules were observed. Nevertheless, in pH = 7.4 buffer incubated group, all intact HCeO_2_ were discovered even after 4 h of co-culture (Figure 3D). In addition, TEM images indicated time-dependent decomposition performance of HCeO_2_@ICG-RGD under acidic microenvironment. Moreover, the released ICG in the supernatant was recorded by UV-*vis* spectra. Almost more than 90% of ICG loading were detected after the pH = 5.6 buffer was treated for 4 h, which is consistent with the released Ce ions content at the same time point (Figure 3E). All results verified the ultrasensitive pH-sensitive degradation capability of HCeO_2_@ICG-RGD, which could be applied for drug precise delivery and improved CDT efficiency, in comparison with the previously reported non-degradable CeO_2_ nanoparticles [37,38,39]. Considering the strong absorption at near-infrared region (NIR) of ICG, the potential photothermal conversion behavior of HCeO_2_@ICG was studied by 808 nm laser irradiation. As shown in Figure 3F,G, the temperature variation of HCeO_2_@ICG displayed a time-dependent property. The results indicated that HCeO_2_@ICG showed a superior photothermal generation performance with ΔT = 22 °C increase after 5 min of irradiation by 808 nm NIR laser irradiation. While the temperature in the HCeO_2_ group presented a slight increase (ΔT = 1 °C) after 5 min of exposure. By comparison with HCeO_2_@ICG, it demonstrated that ICG played a vital role in hyperthermia generation. Furthermore, all photothermal findings implied that HCeO_2_@ICG could be utilized as an effective NIR photothermal agent.

### 3.3. Cellular Uptake, ROS Generation, and Cell Killing Effect

Furthermore, the cell viability of breast tumor cells treated with HCeO_2_@ICG−RGD was evaluated. At different experimental concentrations, HCeO_2_@ICG−RGD had a negligible cytotoxicity toward normal cells (HUVEC) after 24 h of co-culture, and the cell viability still maintained a comparatively high (90%) viability (Appendix A). Interestingly, after incubation with 400 μg/mL of nanoenzymes for 24 h, only 75% of cells were alive (Figure 4A). The data suggested that our CeO_2_-based nanoparticles exhibited excellent biocompatibility in normal cells and demonstrated notable cytotoxicity against tumor cells. This is primarily due to the CDT in CeO_2_-based nanoagents. Subsequently, we studied the tumor cells targeting capability of this nanoenzyme. According to the CLSM images of 4T1 cells after treatment with HCeO_2_@ICG-RGD or HCeO_2_@ICG, the red fluorescence from ICG was higher in the HCeO_2_@ICG-RGD group (Figure 4D). Meanwhile, as shown in Figure 4B, the quantitative fluorescent intensity was calculated, demonstrating that HCeO_2_@ICG-RGD incubated tumor cells possessed significantly higher ICG in comparison with the non-RGD modified group. Various types of cell membrane receptors overexpressed in cancerous cells have been applied for receptor-mediated targeted delivery. One well-known receptor is α_v_β_3_ integrin, which is highly overexpressed in the tumor-associated vasculature and several types of tumor cells, such as 4T1 murine breast cancer cells [40]. All of the above findings proved an efficient α_ν_β_3_ selectivity during breast tumor cell recognition in vivo. Intracellular ROS generation experiment was latterly conducted in cancerous cells. In PBS treatment, the green fluorescence was invisible, while strong fluorescence could be detected in HCeO_2_@ICG-RGD groups, testifying the CDT effect of CeO_2_-based nanoenzymes. Then, the influence of PTT on ROS production was monitored. Surprisingly, compared with the single CDT group with HCeO_2_@ICG-RGD co-culture, we found that more dramatic green fluorescence was detected in the NIR laser irradiated group (Figure 4E). Accordingly, the mean fluorescent intensity of intracellular ROS was prominently higher than PBS and CDT groups (Figure 4C), further confirming that hyperthermia from PTT was beneficial for cytoplasm ROS generation. It has been demonstrated that sufficient hydroxyl radicals generation is capable of causing irreversible damage to organelles, especially for mitochondria, which can block protein transcription and reduce ATP levels that ultimately result in functional protein destruction. Meanwhile, Fenton-like reaction could further improve the tumor sensitivity toward hyperthermia. Therefore, the synergistic cell killing effect estimation was carried out by the CCK-8 kit assay in 4T1 cells under 808 nm laser irradiation. Clearly, in sharp contrast with single CDT with 75% live cells, only 38% cell viability was discovered in HCeO_2_@ICG-RGD+laser group. Meanwhile, as shown in Figure 4F,G, the live/dead cells discrimination of synergistic PTT+CDT had the strongest tumor cell suppression ability in comparison with other formulations. Finally, the cell apoptosis ratio was conducted by Annexin V-FITC/PI assays after 12 h of treatment. Remarkably high apoptosis ratio was found in combination with the treated group (Figure 4H,I). In contrast, single CDT, PBS, and ICG+laser groups did not exhibit apoptotic tendency. Importantly, HCeO_2_@ICG-RGD+laser displayed 45% cell apoptosis/necrosis after 24 h of incubation, which is in agreement with CCK-8 kit evaluation (Appendix A). Furthermore, all of the above cell killing investigations testified the overwhelming cell killing performance of PTT+CDT.

### 3.4. Tumor Targeting and Tumor Inhibition Capability In Vivo

After demonstrating the synergistic therapy function of HCeO_2_@ICG-RGD in tumor cells, we would like to investigate this nanoenzyme in tumor model in vivo. First, we employed second near-infrared region (NIR II) fluorescence imaging (808 nm laser, 400 W cm^−2^, 1000 nm long pass filter) in vivo to trace these nanoenzymes in subcutaneous 4T1 breast cancer bearing-mice after tail vein injection with HCeO_2_@ICG-RGD (8.2 mg/kg) (Figure 5A). Meanwhile, no other NIR II fluorescence signals were observed in tumor site after free ICG injection. The ICG fluorescent signals in the tumor tissue of HCeO_2_@ICG-RGD increased and reached a maximum level at 24 h post-injection, revealing the tumor accumulation capability of those HCeO_2_@ICG-RGD nanoparticles. Meanwhile, ex vivo NIR II fluorescent images of major organs and tumors collected at 24 h post-injection were used for semi-quantitative biodistribution. As shown in Appendix A, the high tumor uptake of nanoenzymes was found in nanoenzyme group, and invisible signals were tracked in ICG group, further illustrating the tumor targeting performance of HCeO_2_@ICG. Accordingly, PTT was performed at this peak accumulation time point. In vivo IR imaging results demonstrated that after irradiation by 808 nm laser for 5 min, the tumor tissue temperature of the control group mice was not notably increased and rapidly heated from 32 to 50 °C. Due to the enhanced permeability and retention (EPR) effect and RGD targeting modification that enhanced the affinity of HCeO_2_@ICG-RGD toward tumor cells, a more efficient enrichment of nanoenzymes in the tumor tissue was observed, indicating that it could be used as an effective photothermal agent in vivo. Next, the tumor multi-mode therapy effect was further performed. As shown in Figure 5C,D, HCeO_2_@ICG-RGD+laser had an excellent synergistic PTT+CDT antitumor ability with the tumor volume clearly reducing after 18 days of treatment. Moreover, the signal CDT administration (HCeO_2_-RGD+laser) demonstrated some excellent tumor suppression capability. Furthermore, RGD modification can markedly improve the anticancer effect of tumor aggregation and augmentation. After 9 days of treatment, tumors were dissected for H&E and TUNEL staining analysis. No apparent apoptosis/necrosis was found in PBS group. Interestingly, HCeO_2_-RGD+laser and HCeO_2_@ICG+laser caused moderate apoptosis/necrosis by tumor targeting enhancement of single CDT and PTT after the EPR effect. Large areas of apoptosis/necrosis were detected in HCeO_2_@ICG-RGD+laser group (808 nm laser 1 W cm^−2^, 5 min) (Figure 5F). The synergistic PTT and CDT induced the most clear cell damage, especially apoptosis, which is also in accordance with tumor growth curves. Meanwhile, there were no apparent changes in body weight and pathological injury major organs, indicating the excellent biocompatibility of HCeO_2_@ICG-RGD in vivo (Figure 5E and Appendix A). Altogether, these data proved that our HCeO_2_@ICG-RGD nanoplatform with excellent biosafety exhibited high potential for the clinical application of enhancing therapeutic efficacy via synergistic CDT/PTT.

## 4. Conclusions

In conclusion, hollow mesoporous CeO_2_-based nanoenzymes with ICG loading and RGD surface anchoring for synergistic CDT+PTT were successfully constructed using a hard-template strategy. The as-made HCeO_2_@ICG-RGD demonstrated biodegradable behavior under acidic microenvironment in cytoplasm for ICG and Ce ions co-release. Then, massive ROS generation from CeO_2_-based nanogranules was found under H_2_O_2_ triggering. The ICG release has an excellent photothermal conversion capability under 808 nm laser irradiation. Furthermore, PTT could effectively assist CDT due to the rise in hyperthermia, and, in turn, CDT made tumor cells more susceptible toward PTT. In vivo results exhibited that our nanoenzymes had excellent tumor targeting capability via active selectivity and EPR effect. More importantly, mutually reinforcing CDT and PTT can effectively eradicate the malignant solid breast tumors without systematic toxicity. This work paved a fascinating way to obtain self-reinforced PTT+CDT with high specificity based on CeO_2_-based nanoplatform, which will have a potential value in cancerous therapy.

Although HCeO_2_@ICG-RGD demonstrated a small amount of off-targeted side effect during the tumor eradication period, the considerations regarding the long-term biosafety should still be seriously considered. Further studies on the biocompatibility and biodistribution of our CeO_2_-based nanoenzymes and their biodegradation byproducts should also be conducted. Owing to the main obstacle of the preclinical translation of nanoparticles is the uncontrollable fate and systematic harmfulness in vivo. Once inoculated, it remains difficult to follow the trail of the fate of nanoparticles in vivo. Additionally, this can allow the targeted tumor site to reach other major organs, where they can extensively accumulate and be toxic. Therefore, investigations on how to fabricate safer CeO_2_-based nanoenzymes in vivo, through functionalized or modification strategies, are very crucial.

## Figures and Tables

**Figure 1 pharmaceutics-14-01717-f001:**
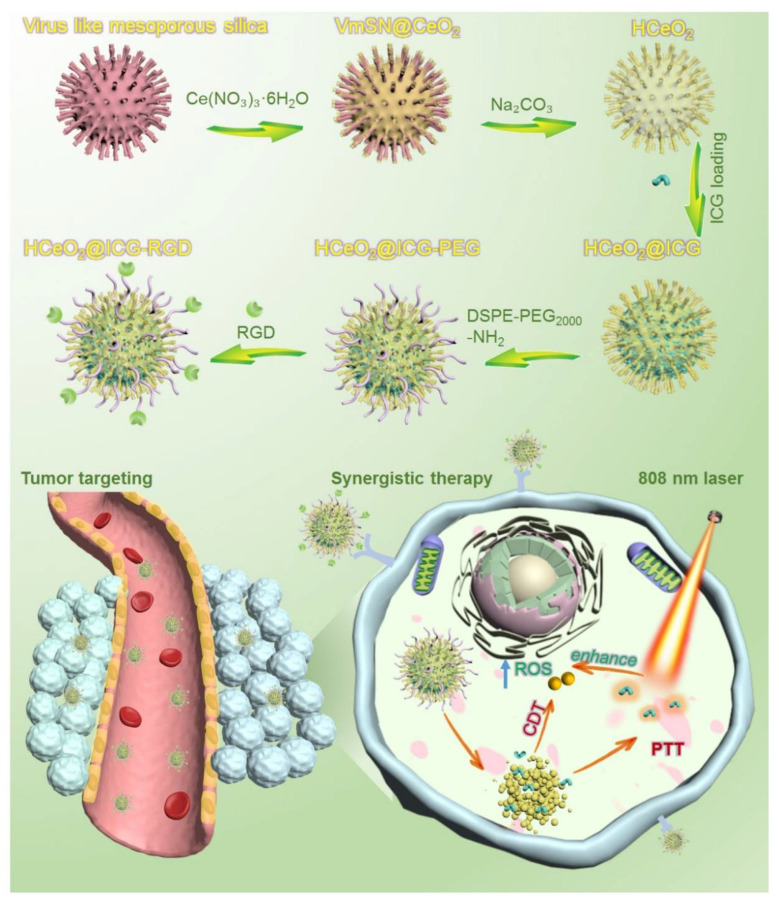
Schematic illustration of stepwise HCeO_2_@ICG-RGD fabrication for precise synergistic eradication of malignant tumors via PTT enhanced CDT.

**Figure 2 pharmaceutics-14-01717-f002:**
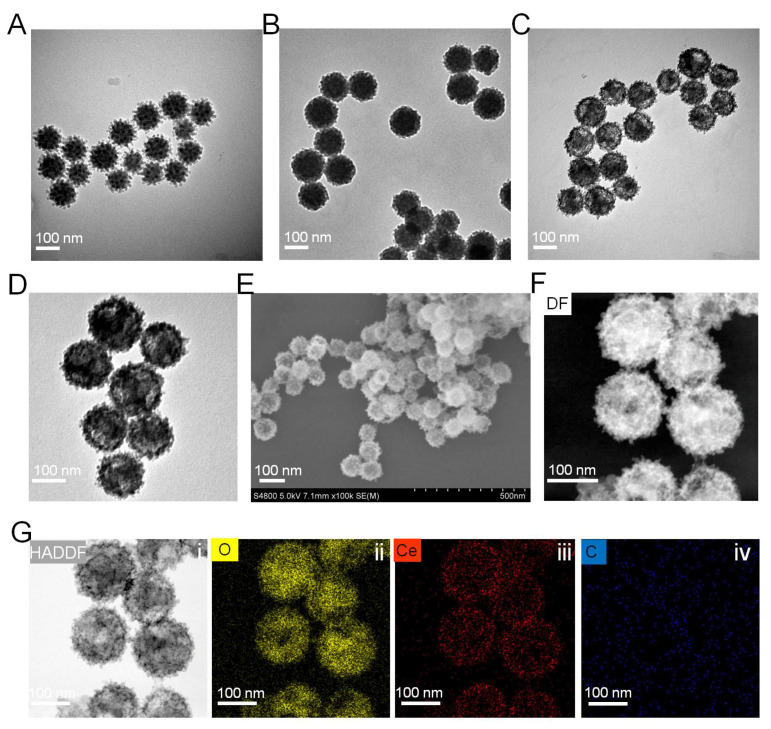
TEM images of virus-like mesoporous silica (**A**), CeO_2_ coated mesoporous silica (**B**), HCeO_2_ nanoparticles (**C**), and HCeO_2_@ICG-RGD (**D**). SEM images of HCeO_2_@ICG-RGD (**E**). Dark field (**F**), HADDF (**Gi**), O (**Gii**), Ce (**Giii**), C (**Giv**) elemental distribution images of HCeO_2_@ICG-RGD.

**Figure 3 pharmaceutics-14-01717-f003:**
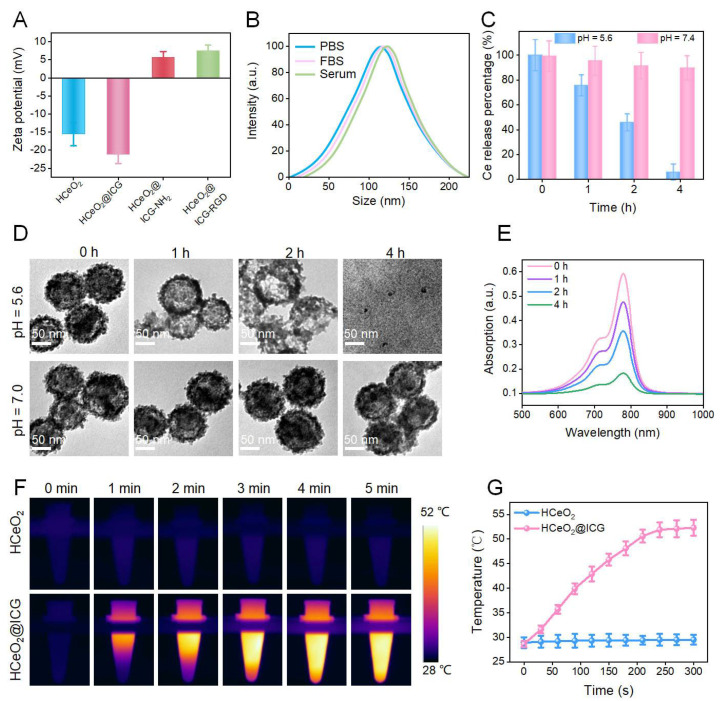
Zeta potential values of HCeO_2_, HCeO_2_@ICG, HCeO_2_@ICG−NH_2_, and HCeO_2_@ICG−RGD (**A**). Size distribution of HCeO_2_@ICG-RGD dispersed in PBS, FBS, and serum for 7 days (**B**). Ce ions release (**C**), TEM images (**D**) of HCeO_2_@ICG−RGD incubated in pH = 5.6 and 7.4 buffers for different times. (**E**)The released ICG fluorescence intensity of HCeO_2_@ICG−RGD incubated in pH = 5.6 for various durations. Infrared thermal images and (**F**) and temperature curve (**G**) of the solution containing HCeO_2_ or HCeO_2_@ICG under the 808 nm laser (0.75 W cm^−2^) irradiation for different minutes.

**Figure 4 pharmaceutics-14-01717-f004:**
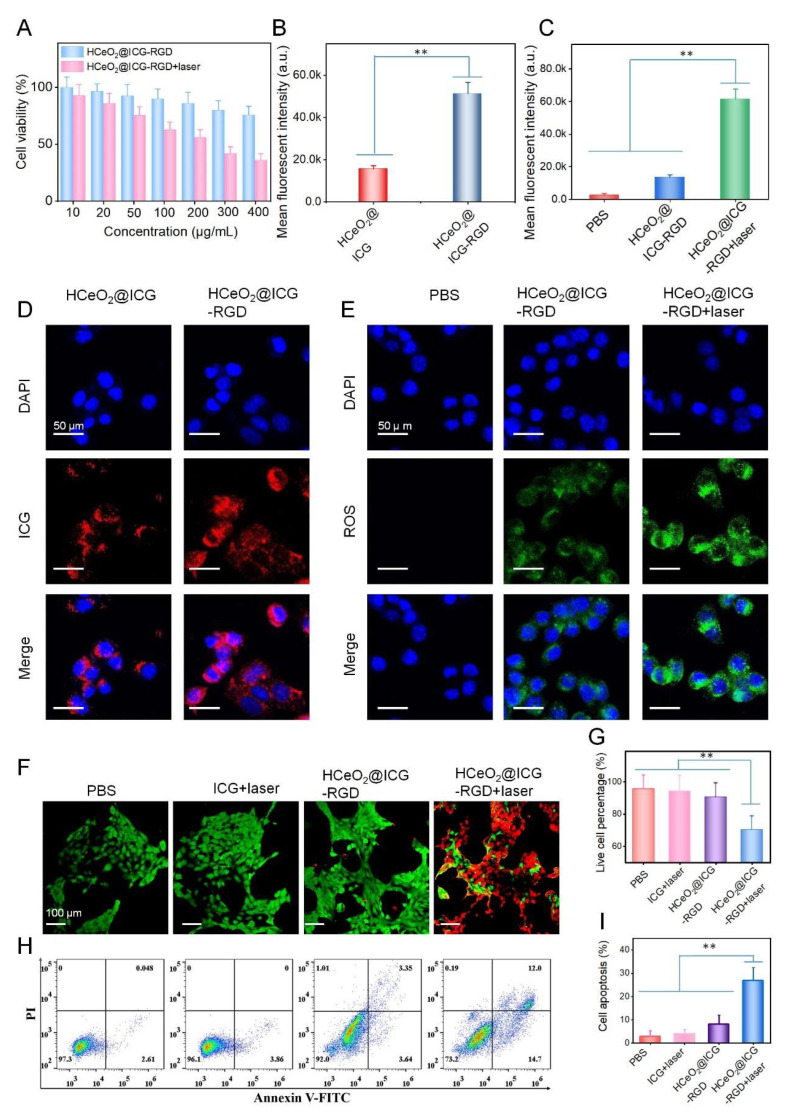
(**A**) Cell viability of 4T1 cells after 12 h of treatment with HCeO_2_@ICG-RGD and HCeO_2_@ICG-RGD+laser (808 nm, 0.75 W cm^−2^, 5 min) at various concentrations. (**B**,**C**) Quantitative fluorescent intensity analysis of CLSM images in (**D**,**E**), respectively. (**D**) CLSM images of 4T1 cells after treatment with HCeO_2_@ICG or HCeO_2_@ICG-RGD for 12 h. (**E**) CLSM images of intracellular ROS generation after treatment with PBS, HCeO_2_@ICG-RGD, and HCeO_2_@ICG-RGD+laser for 12 h. Fluorescent images of live/dead cells (**F**) and flow cytometer analysis of cell apoptosis (**H**) after treatment with PBS, ICG+laser, HCeO_2_@ICG-RGD, and HCeO_2_@ICG-RGD+laser for 6 h. (**G**,**I**) Quantitative analysis of total live cells in (**F**) and cell apoptosis in (**H**), respectively. Mean ± SD for n = 4 (** *p* < 0.01 versus HCeO_2_@ICG-RGD+laser, two-sided Student’s *t*-test).

**Figure 5 pharmaceutics-14-01717-f005:**
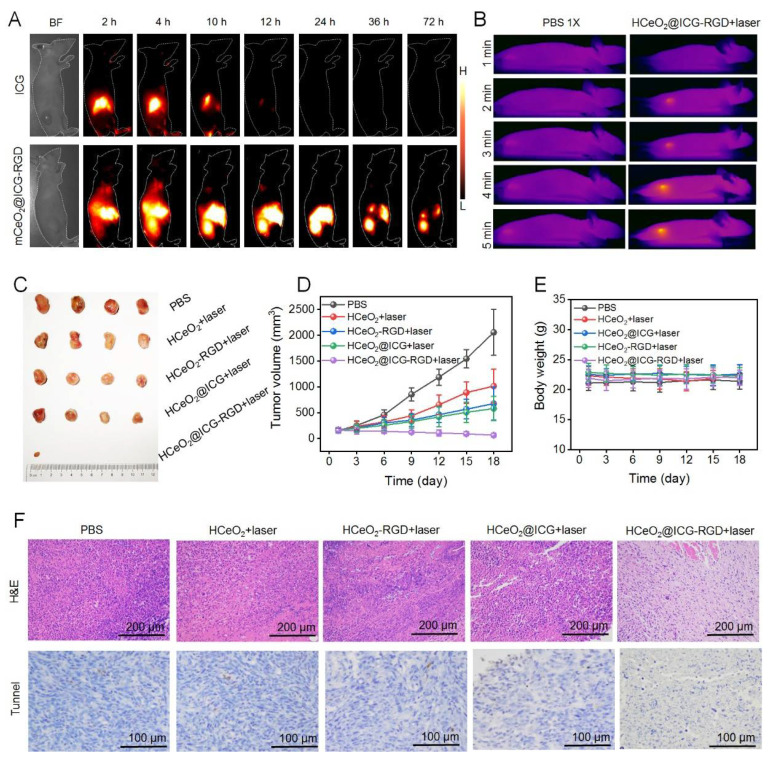
(**A**) NIR II fluorescent images of 4T1 breast tumor-bearing nude mice after tail vein administration of ICG or HCeO_2_@ICG-RGD at various times. (**B**) IR thermal images of 4T1 breast tumor-bearing nude mice under 808 nm laser (1 W cm^−2^) irradiation for different durations. Two groups of BALB/c nude mice were pretreated with tail vein administration of PBS or HCeO_2_@ICG-RGD at 24 h. (**C**) Digital image of dissected solid breast tumors after 18 days of treatment with PBS, ICG+laser, HCeO_2_-RGD+laser, HCeO_2_@ICG+laser, and HCeO_2_@ICG-RGD+laser. Tumor volume curves (**D**) and body weights measurement (**E**) of 4T1 breast tumor-bearing nude mice after PBS, ICG+laser, HCeO_2_-RGD+laser, HCeO_2_@ICG+laser, and HCeO_2_@ICG-RGD+laser administration for different days. (**F**) H&E and TUNEL staining images of tumor tissues after 7 days of administration with PBS, ICG+laser, HCeO_2_-RGD+laser, HCeO_2_@ICG+laser, and HCeO_2_@ICG-RGD+laser.

## Data Availability

Not applicable.

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
