# Peer review of "Hollow Mesoporous CeO2-Based Nanoenzymes Fabrication for Effective Synergistic Eradication of Malignant Breast Cancer via Photothermal–Chemodynamic Therapy"

_pharmaceutics, 2022, doi:10.3390/pharmaceutics14081717_

Round 1

Reviewer 1 Report

Dear Authors,

the article is novel and well develop, but since you are working with silica particles, partciles basic characterization method is missed in this paper.

With regards,

Hollow mesoporous CeO2 based nanoenzymes fabrication for effectively synergistic eradication of malignant breast cancer via photothermal-chemodynamic therapy

1.      The abstract should be rewritten in order to clarify the idea of the article.

2.      Too many abbreviations in the abstract have been found, in some cases we don’t know which is the meaning of the words.

3.      References should be named equal in the whole text (for example reference 33).

4.      Line 109: than12 h

5.      Characterization such as Powder Diffraction X ray, N2 adsorption desorption and IR of the HCeO2, HCeO2@ICG, HCeO2@ICG-NH2 and HCeO2@ICG-RGD spectrum should be developed.

Andrea

Author Response

Reviewer #1:

This work is about synthesis of a hallow mesoporous CeO2 based nanoenzymes for photothermal-chemodynamic therapy. Preclinical model of malignant breast cancer cells was used in this study. This work is topical and quite novel. I only have some minor comments:

Response: Thanks very much for the reviewer’s comments. The arisen issues have been carefully addressed following the reviewer’s suggestion.

Comments 1

Abstract, L17: Please provide the long form of CDT (i.e. chemodynamic therapy).

Response: Thanks very much for the useful comments. We have provided the long form of CDT in the abstract section.

Comments 2

Introduction, L50: For nanomaterial application in cancer therapy, please use some updated references such as Siddique et al (Nanomaterials 2020,10,1700).

Response: Thanks very much for the useful comments.

We have added the following citation in the references section of the revised manuscript.

  1. Siddique, S.; Chow, J.C.L. Application of Nanomaterials in Biomedical Imaging and Cancer Therapy. Nanomaterials 2020, 10, 1700.  [PubMed]

Comments 3

 Introduction: Can the authors use the English name of “̇·OH”?

Response: Thanks very much for the useful comments. We have replaced •OH with its English name, hydroxyl radicals in the introduction section.

Comments 4

Authors can discuss whether the RGD is tumour cell specific (e.g. only for breast cancer cell in this study)?

Response: Thanks very much for the useful comments. Various types of cell membrane receptors overexpressed in cancerous cells have been applied for receptor-mediated targeted delivery. One famous receptor is αvβ3 integrin that is highly overexpressed in the tumor-associated vasculature and several kinds of tumor cells, such as 4T1 murine breast cancer cells.

The relevant description has been added in section 3.3. of the revised manuscript as follows: “Various types of cell membrane receptors overexpressed in cancerous cells have been applied for receptor-mediated targeted delivery. One famous receptor is αvβ3 integrin that is highly overexpressed in the tumor-associated vasculature and several kinds of tumor cells, such as 4T1 murine breast cancer cells. [40] All above findings proved an efficient ανβ3 selectivity during breast tumor cell recognition in vivo.”

We have added the following citation in the references section of the revised manuscript.

  1. Li, H.; Wang, M.; Huang, B.; Zhu, S.W. Zhou, J.J.; Chen, D.R.; Cui, R.; Zhang, M.; Sun, Z.J. Theranostic near-infrared-IIb emitting nanoprobes for promoting immunogenic radiotherapy and abscopal effects against cancer metastasis. Commun. 2021, 12, 7149. [PubMed]

Comments 5

 Figure 2: Please continue label the sub-figures of 2Gi, 2Gii, 2Giii and 2Giv in the figure caption.

Response: Thanks very much for the useful suggestions. We have added the description of sub-figures of 2Gi, 2Gii, 2Giii and 2Giv in the figure caption of the revised manuscript.

Comments 6

Figure 3: “Figure 3G” is missing at the end of the figure caption. Same problem in Figure 5F.

Response: Thanks very much for the useful suggestions. We have added the figure captions of Figure 3G and Figure 5F in the revised manuscript. Meanwhile we have also checked thoroughly the figure captions in the revised manuscript.

Comments 7

  The authors should discuss any future work and next step on their study. They also need to discuss any limitations in their preclinical experiments.

Response: Thanks very much for the useful comments.

The relevant sentences have been added in conclusion section of the revised manuscript as follows: “Even though HCeO2@ICG-RGD presented little off-targeted side effect during tumor eradication period, still the considerations regarding the long term biosafety should be seriously taken into account. Further studies of the biocompatibility and biodistribution of our Ce-based nanoenzymes and their biodegradation byproducts should also be conducted. Owing to the main hurdle to preclinical translation of nanoparticles is the uncontrollable fate and systematic harmfulness in vivo. Once inoculated, it remains difficult to follow the trail of the fate of nanoparticles in vivo. Additionally, they can leave the targeted tumor site reach other major organs where they can extensively accumulate and become toxic. Therefore, investigations on how to fabricate CeO2 nanoenzymes safer in vivo, either through functionalized or modification strategies, are very crucial.”

Reviewer 2 Report

Referee Report

Title: Hollow mesoporous CeO2 based nanoenzymes fabrication for effectively synergistic eradication of malignant breast cancer via photothermal-chemodynamic therapy

Manuscript ID: pharmaceutics-1829941

By Joiner et al

Submitted to Pharmaceutics (ISSN 1999-4923)

Comments

This work is about synthesis of a hallow mesoporous CeO2 based nanoenzymes for photothermal-chemodynamic therapy. Preclinical model of malignant breast cancer cells was used in this study. This work is topical and quite novel. I only have some minor comments:

1.       Abstract, L17: Please provide the long form of CDT (i.e. chemodynamic therapy).

2.       Introduction, L50: For nanomaterial application in cancer therapy, please use some updated references such as Siddique et al (Nanomaterials 2020,10,1700).

3.       Introduction: Can the authors use the English name of “̇·OH”?

4.       Authors can discuss whether the RGD is tumour cell specific (e.g. only for breast cancer cell in this study)?

5.       Figure 2: Please continue label the sub-figures of 2Gi, 2Gii, 2Giii and 2Giv in the figure caption.

6.       Figure 3: “Figure 3G” is missing at the end of the figure caption. Same problem in Figure 5F.

7.       The authors should discuss any future work and next step on their study. They also need to discuss any limitations in their preclinical experiments.

Author Response

Reviewer #2:

The article is novel and well develop, but since you are working with silica particles, partciles basic characterization method is missed in this paper.

Response: Thanks very much for the reviewer’s comments. The arisen issues have been carefully addressed following the reviewer’s suggestion.

Comments 1

The abstract should be rewritten in order to clarify the idea of the article.

Response: Thanks very much for the useful suggestions. We have rewritten the abstract in order to clarify the highlights of the article.

The relevant sentences have been added in abstract section of the revised manuscript as follows: “CeO2 based nanoenzymes present a very promising paradigm in cancerous therapy as H2O2 can be effectively decomposed under the electron transmit between Ce3+ and Ce4+. However, the limitation of endogenous H2O2 and intracellular low Fenton like reaction rate lead to single unsatisfied chemodynamic therapy efficacy. Other therapeutic modalities combine with chemodynamic therapy is generally used to enhance the tumor eradiation efficacy. Here, we have synthesized a novel hollowed pH sensitive CeO2 nanoenzyme, after cavity loaded with Indocyanine green (ICG) and surface modification of tumor targeting peptides, Arg-Gly-Asp (denote as HCeO2@ICG-RGD), it could successfully targeted toward tumor cells via αvβ3 recognition. Importantly, in comparison with single chemodynamic therapy, large amount of reactive oxygen species in cytoplasm was induced via photothermal therapy enhanced chemodynamic therapy. Furthermore, tumor cells were efficiently killed with phtothermal therapy and chemodynamic therapy combination, revealing the synergistic therapy was successfully constructed. This is mainly due to ICG precise delivery and release after HCeO2 decomposition in cytoplasm, then effective hyperthermia generation was found under 808 nm laser illumination. Meanwhile, our HCeO2@ICG-RGD can be acted as a fluorescent imaging contrast agent for evaluate tumor tissues targeting capability in vivo. Finally, we found almost all tumors in HCeO2@ICG-RGD+laser groups were completely eradicated in breast cancer bearing mice, further proving the effectively synergistic effect in vivo. Therefore, our novel CeO2 based PTT photothermal agents provides a proof-of-concept argumentation of tumor-precise multi-mode therapies in preclinical.”

Comments 2

Too many abbreviations in the abstract have been found, in some cases we don’t know which is the meaning of the words.

Response: Thanks very much for the useful suggestions. We have provided all full names of the abbreviations in the abstract of the revised manuscript.

Comments 3

 References should be named equal in the whole text (for example reference 33).

Response: Thanks very much for the useful suggestions. We have provided all references with the same format.

Comments 4

Line 109: than12 h.

Response: Thanks very much for the useful comments. We have corrected this typo.

Comments 5

Characterization such as Powder Diffraction X ray, N2 adsorption desorption and IR of the HCeO2, HCeO2@ICG, HCeO2@ICG-NH2 and HCeO2@ICG-RGD spectrum should be developed.

Response: Thanks very much for the useful suggestions. We have provided the specific surface area, XRD and FTIR spectra of Ce-based nanoenzymes in our revised supporting information.

The relevant sentences of XRD spectra have been added in section 3.1. of the revised manuscript as follows: “Simultaneously, as displayed in X-ray diffraction (XRD) results, after ICG loading and RGD surface modification, a single broad diffused peak was still maintained in HCeO2@ICG-RGD, hinting that the nanogranules packed HCeO2 nanoenzymes are more susceptible to degradation (Figure S2).”

The relevant sentences of N2 absorption spectra have been added in section 3.1. of the revised manuscript as follows: “At the same time, we found the specific surface area of HCeO2@ICG-RGD was decreased to 41.6% of the original HCeO2, resulting from the cavity loading of ICG, surface anchoring of both DSPE-PEG2000-NH2 and RGD (Figure S3).”

The relevant sentences of FTIR spectra have been added in section 3.1. of the revised manuscript as follows: “As shown in Figure S1, the peak of1423 cm-1 in Fourier transform infrared spectra (FTIR) was attributed to -C-N- from ICG and it disappeared, resulting from DSPE-PEG2000-NH2 modification. Interestingly, Due to the formation of amide bond between HCeO2@ICG-NH2 and RGD, it reappeared again obviously.”

We have added Figure S1-S3 in the revised supplementary materials as follows.

Figure S1. Nitrogen adsorption–desorption isotherms of HCeO2 (A) and HCeO2@ICG-RGD (B).

Figure S2. FTIR spectra of HCeO2, HCeO2@ICG, HCeO2@ICG-NH2 and HCeO2@ICG-RGD.

Figure S3. XRD spectra of HCeO2, HCeO2@ICG, HCeO2@ICG-NH2 and HCeO2@ICG-RGD.

Round 2

Reviewer 1 Report

Dear Authors,

thank you very much for the corrections.

In my opinion the article could be published as the current form.

Thank you,

Andrea 

Reviewer 2 Report

I am satisfied with the responses from the authors to my comments, and their corresponding corrections/modifications made by the authors.